# Prevalence of Sexual Dysfunction in Mexican Women with Rheumatoid Arthritis

**DOI:** 10.3390/healthcare10101825

**Published:** 2022-09-21

**Authors:** Wendoline Rojo-Contreras, Valeria Diaz-Rizo, Xochitl Trujillo, Miguel Huerta, Alberto D. Rocha-Muñoz, Benjamin Trujillo-Hernandez, Alicia Rivera-Cameras, Ingrid P. Dávalos-Rodríguez, Mario Salazar-Páramo

**Affiliations:** 1Coordinación Clínica de Educación e Investigación en Salud, Hospital General de Zona No. 14, Instituto Mexicano del Seguro Social (IMSS), Guadalajara 06600, Mexico; 2Departamento de Disciplinas Filosófico, Metodológicas e Instrumentales, Centro Universitario de Ciencias de la Salud (CUCS), Universidad de Guadalajara, Guadalajara 44100, Mexico; 3Centro Universitario de Investigaciones Biomédicas, Universidad de Colima, Colima 28040, Mexico; 4Departamento de Salud-Enfermedad como Proceso Individual, Centro Universitario de Tonalá, Universidad de Guadalajara, Guadalajara 44100, Mexico; 5División de Genética, Centro de Investigación Biomédica de Occidente, IMSS y Doctorado en Genética Humana, CUCS, Universidad de Guadalajara, Guadalajara 44100, Mexico; 6Departamento de Fisiología, CUCS, Universidad de Guadalajara, Guadalajara 44100, Mexico

**Keywords:** female sexual dysfunction, rheumatoid arthritis, quality of life

## Abstract

We estimate the prevalence and identified the associated factors of sexual dysfunction in Mexican women with rheumatoid arthritis (RA). A cross-sectional survey was applied to 100 women with RA and compared with 100 healthy, sexually active, adult women. Assessments included an interview using the Female Sexual Function Index (FSFI). Assessment of factors related to sexual dysfunction included gynecologic characteristics, disease activity (DAS-28), and functioning questionnaire (HAQ-DI). Mann-Whitney U test and the Chi-square test were used to compare medians and proportions between the groups. A multivariate logistic regression was performed using sexual dysfunction according to impairments shown by the FSFI. A higher proportion of RA patients had sexual dysfunction compared with controls. Domains with higher impairment in RA patients were desire, arousal, lubrication, and orgasm. A decrease in sexual function correlated with age (r = −0.365 *p* < 0.001) and higher scores in HAQ-DI (r = −0.261 *p* = 0.009). Those patients with a higher disability had higher impairments in desire, arousal, lubrication, and satisfaction. In the multivariate analysis, menopause was associated with sexual dysfunction (OR: 10.02; 95% CI: 1.05–95.40, *p* = 0.04), whereas use of methotrexate was a protective factor (OR: 0.32; 95% CI: 0.11–0.92, *p* = 0.03). Sexual dysfunction is highly prevalent in Mexican women with RA. Clinicians should systematically evaluate the impairment in sexual function in women with RA.

## 1. Introduction

Rheumatoid arthritis (RA) is a chronic inflammatory rheumatic disease characterized by inflammation and progressive joint damage, its prevalence is between 1–2% in the world population [1,2]. In Mexico, the prevalence of RA varies from 0.7% to 2.8%, according to the different regions [3]. Predominantly affecting women aged between 30 and 50 years, around the world RA can affect many quality-of-life components, including a patient’s sexuality [4,5,6]. Female sexual function (FSF) represents a complex interaction of several components, including anatomic, physiological, medical, and social aspects that are closely related. Disturbances in one or more of these components result in sexual dysfunction [7]. Different studies have established that approximately 46% to 75% of females with RA had sexual dysfunction (FSD) [4,5,6,7,8]. Sexual function could be affected in RA due to pain, stiffness, disability, depression, fatigue, as well as the therapy used [9,10]. In Mexico, FSD in healthy women has been reported at 62.1% [11]. Moreover, a study on Mexican RA patients addresses that sexual reproductive health is important in their general health [12].

The Female Sexual Function Index (FSFI) is a self-reported questionnaire constructed to assess six dimensions of sexual function in women with an active sexual life. This index includes the following dimensions: sexual desire, arousal, lubrication, orgasm, and painful intercourse [13]. The FSFI has been validated and can be used to identify an impairment in sexual functioning that may require treatment [14]. A significant decrement in the FSFI scores in patients with RA, compared with controls, was documented. A multiplicity of variables associated with the disease may interact to produce sexual dysfunction, including disease activity, impaired functional performance, extra-articular involvement, and some medications [15,16]. Sexuality is very important in human life, but relevant studies are small among RA patients and differ between populations around the world. Therefore, it is relevant to investigate those variables that influence sexual function. The aim of this work is to identify the prevalence and factors associated with sexual dysfunction in women with RA using a multivariate approach.

## 2. Materials and Methods

Design: Cross-sectional study. Patients were invited to participate from a rheumatology department in a secondary-care hospital, at the Mexican Institute for Social Security (IMSS) (Guadalajara, Mexico) after being approved by the Hospital’s Ethics Committee.

Subjects: One-hundred adult women with RA, diagnosed according to the 1987 American College of Rheumatology criteria [17], were compared with one-hundred healthy women matched by their range of age, who were enrolled by the department of preventive medicine at the same hospital. The women were included if they were between 18 and 50 years old, self-defined as heterosexual, and were sexually active. Patients with RA and controls were excluded if they were pregnant, had a diagnosis of cervical cancer, or were unable (e.g., lacked the education level) to complete the questionnaires.

### 2.1. Characteristics of the Interview

A structured questionnaire was applied to the women, all interviewed by two trained researchers (WRC, VDR) who evaluated the socio-demographic and gynecological characteristics, aspects of sexual behavior, and marital status assessment of sexual function through the FSFI [13]. Briefly, this is a questionnaire that includes nineteen items to explore six domains of sexual function such as sexual desire, arousal, lubrication, satisfaction, orgasm, and pain during intercourse. Each question has 5 or 6 choices, assigning 0 to 5; each domain score is multiplied by a factor that transforms the maximum possible total to 6 (for desire the factor is 0.6; for arousal and lubrication, 0.3; for orgasm, satisfaction, and pain, 0.4). The end result corresponds to the sum of domains, ranging the full-scale scoring from 2.0 to 36.0. Lower scores indicate higher sexual dysfunction [13,18,19]. In order to establish a cut-off point to determine whether a patient has impaired sexual function, Wiegel, et al. computed a cut-off point of 26.55 for the full-scale score to identify patients that had sexual dysfunction [20]. Different breakpoints in various populations have been used for the appropriate cut-off point, established in each from a different population sample based on the scores that were less than the 25th percentile; these were obtained in the healthy Mexican women who constituted our comparison group. Based on this approach, the clinical cut-off points for sexual dysfunction were <21.5 for the full-scale score, and as follows for each domain score: desire <2.4, arousal <3.0, lubrication <3.9, orgasm <3.6, satisfaction <3.6, and pain during intercourse <3.6.

### 2.2. Clinical Assessment in Patients with RA

On the same day, a rheumatologist (ARM, MSP) evaluated the diverse clinical characteristics of the disease in all RA patients, including disease duration, functioning (Health Assessment Questionnaire Disability Index; HAQ-DI) [21], disease activity assessment (DAS28 score) [22], radiological score in the hands by the Steinbrocker stage [23], and extra-articular involvement. In addition, we recorded prednisone, methotrexate, and azathioprine use, the serum rheumatoid factor (RF) levels, and erythrocyte sedimentation rate (Westergren method).

### 2.3. Statistical Analysis

Quantitative variables were expressed as the median and interquartile range (IQR), qualitative variables as frequencies and percentages. Comparisons of the medians between groups were made using the Mann–Whitney U test. The Chi-square test was used to compare the proportions between the two groups. A subgroup analysis was performed to compare the median between quantitative variables in RA patients with sexual dysfunction versus patients without sexual dysfunction. We made Spearman’s correlations between the domains and the general characteristics of the women with RA. A multivariate logistic regression was performed using sexual dysfunction, according to impairments shown by the FSFI, to evaluate factors associated with each domain. In all the models, the weights of the variables were adjusted for age. Odds ratios (ORs) and their 95% confidence intervals (95% CI) were obtained for each factor included in the model. All of the statistical tests were two-sided, and a *p*-value ≤ 0.05 was considered statistically significant. All statistical analyses were performed using SPSS version 16.0 (Chicago, IL, USA).

## 3. Results

### 3.1. Population Characteristics

Table 1 shows the general characteristics of the population, in addition to the sexual history of the patients with and without RA. We can observe that age was not different in the groups, with the median age for RA patients being 41 years (IQR 12), and for the control group, it was 42 (IQR 12) (*p* = 0.205). However, a difference is observed in the active workers (*p* < 0.001), the women with RA being less active. In addition, an association was found in the variables: use of contraceptives (*p* < 0.001), and intercourses per week >3 (*p* = 0.006); with respect to the group of women with RA, they have a minor frequency of these characteristics.

### 3.2. Sexual Dysfunction (FSFI Survey)

The scores for the different domains and the full-scale FSFI between the RA group and controls showed a higher prevalence of sexual dysfunction in the RA group compared with controls (*p* < 0.001), see Table 2.

The comparison of the characteristics of patients with RA, with presence of sexual dysfunction versus those patients without sexual dysfunction, is shown in Table 3. Characteristics associated with sexual dysfunction in the RA group were: multiparity (*p* = 0.03), menopause (*p* = 0.001), higher HAQ-DI scores (*p* = 0.03); whereas methotrexate use was associated with a lower frequency of sexual dysfunction (*p* = 0.04).

Other variables related with lower scores in each domain of the FSFI from patients with RA were analyzed. We found an association of desire impairment with menopause (*p* < 0.01), higher HAQ-DI (*p* < 0.041), and those not treated with methotrexate (*p* = 0.05). The associated variables for arousal impairment were: lower intercourses per week (*p* < 0.01), menopause (*p* < 0.01), higher HAQ-DI (*p* = 0.02), the presence of Sjögren syndrome (*p* = 0.02), and not using methotrexate (*p* < 0.01). The association to impairment in lubrication was related with fewer intercourses per week (*p* = 0.01), menopause (*p* < 0.01), higher HAQ-DI (*p* = 0.02), Sjögren syndrome (*p* = 0.01), and not using methotrexate in treatment (*p* = 0.01).

Table 4 shows the characteristics that correlated with the global score of sexual dysfunction and the scores in each domain for RA patients evaluated. In the global score, the variables that correlated with each domain were: age with arousal (r = 0.329, *p* = 0.001); lubrication (r = −0.412, *p* ≤ 0.001); satisfaction (r = −0.246, *p* = 0.014) and global score (r = −0.365, *p* = <0.001); HAQ-DI with desire (r = −0.242, *p* = 0.0016); arousal (r = −0.216, *p* =0.032); lubrication (r = 0.250, *p* = 0.013); satisfaction (r = 0.246, *p* = 0.014) and global score (r = 0.261, *p* = 0.009); functional class correlated with satisfaction (r = −0.210, *p* = 0.040).

### 3.3. Multivariate Analysis

The results of the multivariate analysis of characteristics associated with impairment in the FSFI and each component are shown in Table 5. In the logistic regression analysis, the factor associated with the presence of sexual dysfunction according to the global results of the FSFI was menopause (OR: 17.96, 95% CI: 2.10–35.37, *p* = 0.008), whereas, the use of methotrexate was observed as a protective factor (OR: 0.28, 95% CI: 0.10–0.77, *p* = 0.01). For each domain, there were different factors associated with impairment. Impairment in desire was associated with menopause (OR: 8.57, 95% CI: 1.89–38.91, *p* = 0.005) and higher scores on the HAQ-DI (OR: 3.47, 95% CI: 1.10–10.95, *p* = 0.03), whereas methotrexate was a protective factor (OR: 0.27, 95% CI: 0.09–0.86, *p* = 0.03). Menopause was associated with an impairment in arousal (OR: 8.10, 95% CI: 1.57–41.92, *p* = 0.01), whereas methotrexate still acted as a protective element (OR: 0.22, 95% CI: 0.08–0.59, *p* = 0.003). Impairment in lubrication was associated with a diagnosis of secondary Sjögren syndrome (OR: 5.84, 95% CI: 1.39–24.61, *p* = 0.02), methotrexate was again a protective factor (OR: 0.23, 95% CI: 0.08–0.68, *p* = 0.008). The domain of satisfaction was associated with menopause (OR: 6.10, 95% CI: 1.57–24.56, *p* = 0.01) and the use of methotrexate was protective (OR: 0.16, 95% CI: 0.06–0.49, *p* = 0.001). Impairment in orgasm only was associated with the patient’s age (OR: 1.09, 95% CI: 1.02–1.17, *p* = 0.008). Finally, an impairment in pain was associated with menopause (OR: 7.05, 95% CI: 1.72–28.81, *p* = 0.007).

## 4. Discussion

In this study, we estimate the prevalence and identified the factors associated with sexual dysfunction (SD) in Mexican women with RA. Previous studies have reported SD as more common among patients with RA compared to controls [5]. We observed that SD was highly prevalent in our studied population and all the domains evaluated were significantly affected. The domains most affected were lubrication, arousal, orgasm, pain during intercourse, satisfaction, and desire. Shahar, et al., also found differences in their results, observing lubrication problems in 17.6%, poor arousal in 21.6%, orgasmic disorder in 7.8%, and reported sexual pain disorder in 19.6%. Additionally, 49% of women did not obtain satisfaction and 31.4% had low desire [24]. SD is dependent on multiple causes that interact to produce the impairment. In this context, this study analyses a wide range of factors (epidemiological, gynecological, and factors specific to the disease) that may influence the index score to evaluate sexual dysfunction in RA. In our study, we used the FSFI as an instrument to identify the presence of sexual dysfunction because the FSFI has been shown to be reliable, have discriminate and divergent validity to distinguish between groups, has a sensitivity to change, and is easy to administer requiring only about 15 min to complete [18,24]. The prevalence of sexual dysfunction among women with RA was evaluated in different independent studies [5,15,24,25,26,27,28]. Particularly, one of them observed that women with RA had a lower global score compared with controls [15]. A systematic review and meta-analysis included studies that evidenced patients with RA have a significantly increased risk of SD, suggesting that both patients and clinicians should be aware of the potential role of RA in the development of SD [29]

Thus in this study, we confirmed a high prevalence (49%) of SD in our patients with RA, similar to the results reported by Orzua-de la Fuente, et al. (51.9%) [30]. Both studies are significantly different in comparison to the study reported by Sahar et al., who observed a frequency of sexual dysfunction in 29.4% of Malaysian patients with RA, although, unfortunately, they did not include a comparison group [23]. Instead, in a study with Egyptian women, SD (with the FSFI) was reported in 45.7% of their patients with RA [25], a value similar to that observed in our study of Mexican women with RA. Factors associated with decreases in the FSFI were observed, showing a relationship between SD and menopause and a higher HAQ-DI score, although there was no relationship with the DAS28 in similarity with Aras, et al. [26]. These findings are different from those reported by others, who observed a correlation between the pain score and disease activity with SD [25], whereas these factors were not associated with the global index score in our patients. The difference in the findings observed among our RA patients could be related to their age, frequency of sexual intercourse, chronicity of symptoms, or the absence of acute disease during the performance of FSFI. The limitation in the performance of sexual intercourse was also observed using the HAQ-DI, identifying that 62% of RA patients had some difficulties, and 17% were completely unable to perform sexual intercourse due to arthritis [4]. Similar to our results, other studies have found a correlation between impairment in the HAQ-DI and sexual disability [4]. Likewise, this association between poor functioning and sexual dysfunction was also reported by Yilmaz et al., who observed a moderate negative correlation between the total FSFI score and the HAQ-DI [15].

In our experience, the HAQ-DI had a negative correlation with sexual parameters including impairments in desire, arousal, lubrication, satisfaction, and global score, whereas functional sexual class was negatively correlated with impaired satisfaction. A difference between our study and two other studies [14,24] is that we found a negative correlation between the prednisone dosage and the FSFI score in two domains, satisfaction and pain, whereas both referenced studies did not evaluate this correlation. On the other hand, one of them observed a negative correlation between the FSFI and DAS28 score, whereas we did not observe this correlation [14]. A possible reason to explain these differences is that their patients had higher disease activity in comparison with our RA patients; therefore, the effect of disease activity on the decrease in sexual function could be influenced by other factors.

Few studies are available for evaluating the factors associated with impaired sexual dysfunction. El-Miedany observed an association between sexual dysfunction and cardiovascular disease, pain score, hip joint involvement, disease activity, tender joint count, and the presence of secondary Sjögren syndrome [25]. We also observed an association between Sjögren syndrome and sexual dysfunction. In a mailed interview using a short version of the Questionnaire for Screening of Sexual Dysfunctions, van Berlo, et al. found that 51% of women with RA and/or Sjögren syndrome had interference with sexual activity caused by pain in their joints [31]. Differences in the methodology and instruments used to evaluate sexual function make it difficult to compare their findings with those of other studies. Similar to our results, others have not found a significant positive association between sexual dysfunction and DMARDs therapy and/or oral steroid therapy [25]. The significance of our study is based on the findings in the analysis, adjusted for age-associated female sexual dysfunction, of factors that were not analyzed in previous studies [15,24,25,26,27]; menopause was associated with global dysfunction and female sexual dysfunction in the domains of desire, arousal, and pain. We observed that sexual dysfunction is highly prevalent in women with no rheumatic disorders. Data produced from a national survey in the United States showed that women experience a loss of sexual interest and almost one-fourth do not experience orgasm [32]. One factor that could be associated with impaired FSFI score is age, as this was described in non-rheumatic populations [33]. Sexual dysfunction may diminish the competence of a patient to achieve satisfactory sexual intercourse with their partner, decreasing the quality of the relationship, and in extreme cases leading to marital unhappiness [34,35,36]. In several studies, about 50–95% of RA patients experienced problems during sexual intercourse, [27,28,33,34]. Likewise, one author reported a high prevalence of depression in women with RA who have sexual dysfunction compared with controls [15].

### Study Limitation

Sexual distress is required to diagnose a disorder of sexual dysfunction. No measure of sexual distress was included, so the impairments in sexual function that we found do not necessarily indicate a full-fledged disorder of sexual function.

## 5. Conclusions

Sexual dysfunction is highly prevalent in Mexican women with RA, and this affected all of the domains evaluated. The factors associated with sexual dysfunction include menopause and a higher score on the HAQ-DI. More effort is required to establish a systematic assessment of this entity and its impact on quality of life.

## Figures and Tables

**Table 1 healthcare-10-01825-t001:** General characteristics and sexual history of patients with rheumatoid arthritis and controls.

Variable	RA *n* = 100	Controls *n* = 100	*p* Value
Age (years), median (IQR)	41 (12)	42(12)	0.205
Married or cohabiting, *n* (%)	82 (82)	82 (82)	1.00
Active workers, *n* (%)	19 (19)	35 (35)	<0.001
Menarche age (years), median (IQR)	12 (3)	12 (2)	0.59
Oral contraceptive use, *n* (%)	3 (3)	63 (63)	<0.001
Oral contraceptives (months), median (IQR)	24 (61)	24(54)	0.504
Age at first intercourse (years), median (IQR)	18 (3)	20 (6)	<0.001
Number of male sexual partners, median (IQR)	1 (1)	1 (2)	0.302
Intercourses per week > 3, *n* (%)	17 (17)	34 (34)	0.006
Multiparity ≥3, *n* (%)	79 (79)	79(79)	1
History of sexually transmitted infections, *n* (%)	7 (7)	2 (2)	0.085
DAS-28, median (IQR)	4.89 (1.04)	-	NC
ESR, median (IQR)	27 (12.8)	-	NC
RF, median (IQR)	18 (130)	-	NC

RA: Rheumatoid arthritis. DAS: Disease activity score. ESR: Erythrocyte sedimentation rate. RF: Rheumatoid factor. IQR: Interquartile range. Comparisons in median were made using the Mann–Whitney U test. Comparisons in proportions were made using the Chi-square test.

**Table 2 healthcare-10-01825-t002:** Comparison in FSFI score and frequency of sexual dysfunction between patients and controls.

Domain	RA *n* = 100	Controls *n* = 100	*p*
Presence of sexual dysfunction (global score <21.5), *n* (%)	49 (49)	25 (25)	<0.001
Desire, median (IQR)	2.4(1.2)	3 (1.7)	<0.001
Impairment in desire (<2.4), *n* (%)	30 (30)	14 (14)	0.006
Arousal, median (IQR)	2.8 (3.9)	4.2 (1.8)	<0.001
Impairment in arousal (<3.6), *n* (%)	50 (50)	23 (23)	<0.001
Lubrication, median (IQR)	3.6 (5.4)	5.1 (2.1)	<0.001
Impairment in lubrication (<3.9), *n* (%)	52 (52)	24 (24)	<0.001
Orgasm, median (IQR)	3.6 (4.7)	4.8 (2.4)	<0.001
Impairment in orgasm (<3.6), *n* (%)	49 (49)	23 (23)	<0.001
Satisfaction, median (IQR)	4 (2.2)	4.8 (2.4)	0.014
Impairment in satisfaction (<3.6), *n* (%)	32 (32)	21 (21)	0.08
Pain intercourse, median (IQR)	4.4 (5.6)	4.8 (2.4)	0.003
Impairment in pain intercourse (<3.6), *n* (%)	36 (36)	20 (20)	0.01
Global score, median (IQR)	21.5 (20.5)	27.2 (9.2)	<0.001

FSFI: Female Sexual Function Index. RA: Rheumatoid arthritis. IQR: Interquartile range. Low FSFI scores indicate impairment in sexual function. Range: For the global index, the scores range from 1.2 to 36 points. Impairment in each domain was computed as a score lower than 25th percentile for normal controls in our population. Comparisons in medians were made using the Mann–Whitney U test. Comparisons in proportions were made using the Chi-square test.

**Table 3 healthcare-10-01825-t003:** Characteristics associated with sexual dysfunction in patients with rheumatoid arthritis.

Characteristics	Sexual Dysfunction *n* = 49	Without Sexual Dysfunction *n* = 51	*p*
Age (years), median (IQR)	45 (6)	40 (14)	0.003
Married or cohabiting, *n* (%)	37 (76)	45 (88)	0.098
Intercourses per week, median (IQR)	1 (2)	2 (2)	0.002
Multiparity ≥3, *n* (%)	43 (88)	36 (71)	0.03
Oral contraceptives use, *n* (%)	2 (4)	1 (2)	0.52
Menopause, *n* (%)	14 (31)	2 (4)	0.001
Comorbidity, *n* (%)	30 (61)	34 (67)	0.57
Disease duration (years), median (IQR)	5(9)	6 (6.7)	0.915
Morning stiffness, median (IQR)	50 (51)	42 (48)	0.323
Severity of pain, median (IQR)	52 (51)	45 (61)	0.309
HAQ-DI median (IQR)	0.77 (0.67)	0.44 (0.71)	0.041
DAS 28, median (IQR)	4.8 (1.09)	4.9 (1.41)	0.879
Sjögren syndrome, *n* (%)	17 (36)	11 (22)	0.12
Rheumatoid nodules, *n* (%)	13 (28)	7 (15)	0.10
Positive rheumatoid factor, *n* (%)	29 (63)	31 (64)	0.87
Prednisone use, *n* (%)	31 (70)	33 (73)	0.76
Methotrexate use, *n* (%)	23 (52)	33 (73)	0.04

Morning stiffness, severity of pain and disease activity were evaluated with visual analogue scale from 0–100 mm. HAQ-DI score ranges 0 to 3. HAQ-DI: Health Assessment Questionnaire Disability Index. DAS28: Disease activity score. IQR interquartile range. Comparisons between median were made using Mann Whitney U-test. Comparisons between proportions were made with the Chi-square test.

**Table 4 healthcare-10-01825-t004:** Correlations between characteristics associated with sexual dysfunction in patients with rheumatoid arthritis.

Characteristic	Desire		Arousal		Lubrication		Satisfaction		Orgasm		Pain		Global Score	
	rho	*p*	rho	*p*	rho	*p*	rho	*p*	rho	*p*	rho	*p*	rho	*p*
Age	−0.170	0.092	−0.329	0.001	−0.412	<0.001	−0.283	0.004	−0.368	<0.001	−0.261	0.009	−0.365	<0.001
HAQ-DI	−0.242	0.016	−0.216	0.036	−0.250	0.013	−0.246	0.014	−0.190	0.061	−0.158	0.121	−0.261	0.009
Functional Class	−0.179	0.081	−0.123	0.234	−0.167	0.104	−0.210	0.040	−0.122	0.237	−0.168	0.101	−0.170	0.097
DAS28	−0.072	0.479	−0.100	0.322	−0.112	0.266	−0.133	0.186	−0.113	0.263	−0.078	0.443	−0.123	0.221
Morning stiffness (last week)	−0.097	0.344	−0.066	0.518	−0.101	0.324	−0.089	0.381	−0.026	0.801	−0.106	0.299	−0.093	0.362
Disease activity (last week)	−0.085	0.407	−0.100	0.329	−0.143	0.161	−0.151	0.137	−0.089	0.383	−0.159	0.117	−0.129	0.204
Prednisone doses	0.013	0.906	−0.108	0.289	−0.092	0.416	−0.113	0.315	−0.060	0.593	−0.123	0.272	−0.068	0.544
Methotrexate doses	−0.013	0.906	−0.054	0.632	−0.113	0.319	−0.080	0.481	−0.093	0.410	−0.122	0.281	−0.086	0.447

HAQ-DI: Health Assessment Questionnaire Disability Index. DAS28: Disease activity score. Bivariate correlations, Spearman´s correlations.

**Table 5 healthcare-10-01825-t005:** Adjusted analysis of factors associated with the presence of sexual dysfunction in rheumatoid arthritis.

Variable	*Method Enter*			*Method Forward Stepwise*		
	*OR*	*95% CI*	*p*	*OR*	*95% CI*	*p*
*Global score*						
Age (years)	1.05	0.98–1.13	0.20	Not in the model		
Menopause	10.02	1.05–95.40	0.04	17.96	2.10–35.37	0.008
HAQ–DI	2.07	0.73–5.91	0.17	Not in the model		
Sjögren syndrome	1.65	0.46–5.95	0.43	Not in the model		
Methotrexate use	0.32	0.11–0.92	0.03	0.28	0.10–0.77	0.01
*Impairment in desire*						
Age (years)	1.04	1.00–1.13	0.31	Not in the model		
Menopause	4.50	1.00–20.11	0.05	8.57	1.89–38.91	0.005
HAQ–DI	3.51	1.14–10.81	0.03	3.47	1.10–10.95	0.03
Methotrexate use	0.33	0.10–0.90	0.05	0.27	0.09–0.86	0.03
*Impairment in arousal*						
Age (years)	1.02	1.00–1.10	0.64	Not in the model		
Menopause	7.09	1.21–41.23	0.03	8.10	1.57–41.92	0.01
HAQ–DI	2.29	0.81–6.45	1.19	Not in the model		
Methotrexate use	0.23	0.08–0.64	0.005	0.22	0.08–0.59	0.003
*Impairment in lubrication*						
Age (years)	1.03	0.96–1.12	0.37	Not in the model		
Menopause	6.50	0.67–62.68	0.11	Not in the model		
HAQ–DI	2.43	0.78–7.56	0.13	Not in the model		
Sjögren syndrome	5.61	1.25–25.11	0.02	5.84	1.39–24.61	0.02
Methotrexate use	0.24	0.08–0.73	0.01	0.23	0.08–0.68	0.008
*Impairment in satisfaction*						
Age (years)	1.07	1.00–1.16	0.12	Not in the model		
Menopause	3.65	0.80–16.63	0.09	6.10	1.57–24.56	0.01
HAQ–DI	1.62	0.56–4.72	0.37	Not in the model		
Methotrexate use	0.19	0.06–0.57	0.003	0.16	0.06–0.49	0.001
*Impairment in orgasm*						
Age (years)	1.07	1.00–1.14	0.06	1.09	1.02–1.17	0.008
Menopause	2.55	0.55–11.75	0.23	Not in the model		
HAQ–DI	1.87	0.70–4.98	0.21	Not in the model		
Methotrexate use	0.40	0.15–1.07	0.07	Not in the model		
*Impairment in pain*						
Age (years)	1.03	0.96–1.11	0.38	Not in the model		
Menopause	4.24	1.03–17.47	0.04	7.05	1.72–28.81	0.007
HAQ–DI	1.51	0.58–3.96	0.40	Not in the model		
Sjögren syndrome	2.40	0.75–7.64	0.14	Not in the model		
Methotrexate use	0.59	0.22–1.61	0.30	Not in the model		

HAQ-DI: Health Assessment Questionnaire Disability Index. OR: Odds ratio. CI: 95% Confidence intervals. Analysis was performed using logistic regression. The dependent variable was sexual dysfunction (yes/no), or impairment in each domain. Statistical significance, *p* ≤ 0.05.

## Data Availability

Data that support the findings of the study are available upon reasonable request.

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
