# Peer review of "Prevalence of Sexual Dysfunction in Mexican Women with Rheumatoid Arthritis"

_healthcare, 2022, doi:10.3390/healthcare10101825_

Round 1

Reviewer 1 Report

Your paper showed original data and conclusions. Please check again the spelling and topic

Author Response

Dear Reviewer,

we really appreciate your comment. We will follow your recommendation, checking spelling and topic.

Best regards,

Authors.

Reviewer 2 Report

Thank you for this well-designed survey study, which circumvents the problem that FSD may be highly prevalent even in women without RA. A number of sentences are phrased in a fashion that require clarification, and one point needs to be added to study limitations. 

Intro

Line 40 “can affect multiplicity …” à “can affect many…”

Line 59 “still being” – this clause is set up to be independent but it lacks a verb. How about something like “Sexual is very important in human life, but relevant studies are small….”

Materials

Line 72 “at the same hospital who” doesn’t make sense. At the same hospital where they participated in a pap smear study?

Line 75 cancer, OR WERE psychologically …

Line 81 …is a questionnaire [no comma] that includes 19 [no hyphen] itemS to …

Line 82 such as SEXUAL desire, arousal…

Line 83 … assigning a score of 0 to 5; each domain score is multiplied by a factor that transforms the maximum possible total to 6 (for desire, the factor is 0.6; for arousal and lubrication, 0.3; for orgasm, satisfaction, and pain, 0.4).

Delete: , and higher scores indicate less sexual dysfunction

Results

Line 153  …were analyzed, was interesting to find an.. à were analyzed. We found an association of desire impairment with…

Discussion

Line 255  …have not found a significant POSITIVE association (we found a negative association, which seems logical) ???

Study limitations

Add: Sexual distress is required to diagnose a disorder of sexual dysfunction. No measure of sexual distress was included, so the impairments in sexual function that we found do not necessarily indicate a full-fledged disorder of sexual function.

Author Response

Dear Reviewer,

thank you for your constructive comments, phrases were clarified as you recommended; also we added a study limitation phrase.

Point 1. Introduction

Line 40 "can affect multiplicity" a "can affect many..."

Response: we change in line 40 the phrase, to "can affect many..."

Line 59 "still being- this clause is set up to be independent but it lacks a verb. How about something like "Sexual is very important in human life but relevant studies are small"

Response: we take your recommendation in line 59 as follows, "Sexuality is very important in human life but relevant studies are small..."

Point 2. Materials

Line 72 "at the same hospital who", doesn´t make sense. At the same hospital where they participated in a pap smear study?

Response: we deleted the phrase "who attended a Pap smear study". 

Line 75 cancer, or were psychologically....

Response: we deleted "psychologically or..."

Line 81 ..is a questionnaire (no comma) that includes 19 (no hyphen) items to..

Response: we deleted comma after questionnaire and hyphen after 19

Line 82 such as sexual, desire, arousal....

Response: we deleted women and incorporated "sexual desire, arousal..."

Line 83 assigning a score of 0 to 5; each domain score is multiplied by a factor that transforms the maximum possible total to 6 (for desire, the factor is 0.6; for arousal and lubrication, 0.3; for orgasm, satisfaction, and pain, 0-4)

Response: lines 81 to 84 were modified, as follows, "choises assigning 0 to 5; each domain score is multiplied by a factor that transforms the maximum possible total to 6, (for desire the factor is 0.6; for arousal and lubrication, 0.3; for orgasm, satisfaction and pain 0.4)".

Delete: and higher scores indicate less sexual dysfunction

Response: phrase was deleted. Line 85. 

Point 3. Results

Line 153 ... were analyzed, was interesting to find  an, to "were analyzed. We found an association of desire impairment with..."

Response: we change the phrase to "were analyzed. We found an association of desire impairment with..."

Line 255 ... have not found a significant POSITIVE association (we found a negative association. ...)

Response: we modified the short phrase adding "positive association"

Study limitations

Add: Sexual distress is required to diagnose a disorder of sexual dysfunction. No measure of sexual distress was included, so the impairments in sexual function that we found do not necessarily indicate a full-fledged disorder of sexual function.

Response: thank you so much for this recommendation, we added the complete paragraph, in substitution of previously.

Again thank you so much.

Reviewer 3 Report

1. The demographic data (age, DAS28, ESR, RF, medication...) of the patients with RA should be added in table 1.

2.Did RA patients received biologic agent? Did biologic agent affect the score of FSFI?

3.Suggest included the reference "J Rheumatol. 2018 Oct;45(10):1375-1382." in the discussion.

4.Disease activity measurement DAS-28 did not correlated with the FSFI. Many studies showed that higher RA disease activity associated with the development of sexual dysfunction. The PI should discuss about this point.   

5. There is a study investigating the sexual dysfunction and its determinants in Moroccan women with rheumatoid arthritis. Pan Afr Med J. 2016; 24: 16. The PI should mention it in the background.

Author Response

Dear Reviewer,

thank you for your comments and helpful suggestions. We included two new references to our manuscript, as it was suggested.

Point 1. The demographic data (age, DAS28 ESR, RF, medication...) of the patients with RA should be added in Table 1.

Response: The demographic, clinimetric and lab data in the whole RA group were added, in Table 1.

Point 2. Did RA patients received biologic agents? Did biologic agent affect the score of FSFI?

Response: Only 5 of our patients received a biologic DMARD, for indetermined time. We do not consider that this small number could affect the FSFI score.

Point 3. Suggest included the reference J Rheumatol 2018 (Oct); 45 (10):1375-1382, in the discussion.

Response: We have included a comment of this reference in discussion section 1st paragraph, lines 215-217. Reference is now number 29

Point 4. Disease activity measurement DAS-28 did not correlate with the FSFI. Many studies showed that higher RA disease activity associated with the development of sexual dysfunction. The PI should discuss about this point.

Response: We have added a short comment in discussion, 2nd paragraph lines 230-232. Textually " The difference in the findings observed among our RA patients could be related to the age, frequency of sexual intercourse, chronicity of symptoms and the absence of acute disease during the performance of FSFI"

Point 5. There is a study investigating the sexual dysfunction and its determinants in Moroccan women with RA. Pan Afr Med J. 2016; 24: 16. The PI should mention it in the background.

Response: We already mentioned the reference from Pan Afr Med J (no. 5). We added in line 40  "around the world" Also,  the new reference is cited in line 211, discussion section. All references were updated.

Thank you.

Reviewer 4 Report

Despite the interesting topic covered in the proposed article on sexual dysfunction and RA, the results presented (unfortunately, only in the form of tables) are very weak, as they largely do not agree with the basic rules of statistical analysis. Therefore, the scientific value of the manuscript in its present form is quite low, since in most cases the incorrect analysis of the data and thus the presentation and evaluation of the results mislead the reader.

Author Response

Dear Reviewer,

thank you for your time and work in reviewing our manuscript.

Point 1. Despite the interesting topic covered in the proposed article on SD and RA, the results presented (unfortunately, only in the form of tables) are very weak, as they largely do not agree with the basic rules of statistical analysis. Therefore, the scientific value of the manuscript in the present form is quite low, since in most cases the incorrect  analysis of the data and thus the presentation and evaluation of the results mislead the reader.

Response: We really appreciate your valuable feedback on our work. After an exhaustive analysis, discussion and a reconsideration of the applied statistical analysis, we conclude that due to the parametric distribution of the data demonstrated with the Kolmogorov-Smirnov test, we used media and standard deviation in order to describe quantitative variables and for qualitative variables frequencies, and percentages. Database could be available upon request.

Best regards.

Round 2

Reviewer 4 Report

Dear authors!

You reffered to non parametric Kolmogorov–Smirnov test to assess normality, however according to the available literaturethe Kolmogorov–Smirnov testshould no longer be used owing to its low power. It is preferable that normality be assessed both visually and through normality tests, of which the D’Agostino-Pearson or the Shapiro-Wilk test are highly recommended.

You still do not want to accept the basic rules of statistics and correctly present your results (detailed explanations of the weakpoints of this manuscript can be found in my previously submitted critical comments). 

You still did not correct results marked in red (see my comments, Table 1, 2, 3), where obviously is seen that these data are not normally distributed and therefore only median with IQR instead of mean+-SD is applicable. 

You still don't want to switch on your logical thinking and to understand that presented results, e.g., in table 1 about oral contraceptives, are nonsense.

Conclusion

Current version of manuscript is not acceptable for publishing in this kind. Only after major revision according previously sent critical comments, the manuscript could be revised again for final decision. 
